# Prevalence and Impact on Quality of Life of Small Intestinal Bacterial Overgrowth (SIBO)-Related Symptoms in Patients with Upper Gastrointestinal Cancer

**DOI:** 10.3390/diseases13120398

**Published:** 2025-12-13

**Authors:** Rosa Rosania, Achim J. Kaasch, Katrin Bose, Friedrich Sinner, Christian Müller, Jochen Weigt, Verena Keitel, Marino Venerito

**Affiliations:** 1Department of Gastroenterology, Hepatology and Infectious Disease, Otto von Guericke University Hospital, 39120 Magdeburg, Germany; 2Faculty of Medicine, Institute of Medical Microbiology and Hospital Hygiene, Otto-Von Guericke University Magdeburg, 39120 Magdeburg, Germany

**Keywords:** small intestinal bacterial overgrowth, upper gastrointestinal cancer, quality of life

## Abstract

Introduction: Although patients with upper gastrointestinal (GI) cancer have an increased risk of developing small intestinal bacterial overgrowth (SIBO) due to disease- and treatment-related factors, SIBO remains underdiagnosed in oncology. Aim and Methods: This prospective study evaluated the prevalence of SIBO and its impact on symptom-related quality of life (QoL) in patients with current or prior upper GI cancer. Between April 2021 and May 2022, patients reporting SIBO-related symptoms like bloating and/or diarrhea completed a standardized symptom questionnaire. QoL impact was scored from 0 (none) to 3 (severe). Patients with scores > 1 and no recent antibiotic use underwent upper endoscopy with duodenal aspirate. SIBO was defined as >10^3^ CFU/mL. Results: Ninety patients were enrolled (51% female; median age of 65 years): 35% had pancreatic, 34% gastric, 17% biliary, and 14% esophageal cancer. Sixty reported SIBO-related symptoms: 35% reported bloating, 11% diarrhea, and 54% both. Of these, 36 underwent endoscopy; 53% were diagnosed with SIBO. Among SIBO-positive patients, 95% reported bloating and 58% reported diarrhea. Prior abdominal surgery was recorded in 63% of SIBO cases. Conclusions: SIBO was identified in more than half of symptomatic upper GI cancer patients, with a strong association with bloating and previous abdominal surgery. These findings emphasize the importance of clinical awareness and appropriate diagnostic evaluation for SIBO in this high-risk group to improve symptom control and quality of life.

## 1. Introduction

Small intestinal bacterial overgrowth (SIBO) is a clinical condition in which gastrointestinal symptoms or laboratory abnormalities result from an excessive or altered bacterial population in the small intestine [1]. The North American consensus defines SIBO as a bacterial colony count of ≥10^3^ colony-forming units per milliliter (CFU/mL) in a duodenal or jejunal aspirate [2].

Under normal physiological conditions, several protective mechanisms, such as coordinated intestinal motility, gastric acid secretion, pancreaticobiliary secretions, and local immune responses, maintain a balanced intestinal ecosystem. Disruption of these mechanisms, due to surgical alteration of anatomy (e.g., gastrectomy or Whipple surgery), chronic opiate use, or hypochlorhydria from long-term proton pump inhibitor (PPI) therapy, predisposes to bacterial overgrowth [3,4].

The exact prevalence of SIBO is not known, but it is believed to be a common clinical disorder. Identifying SIBO clinically is challenging, as no single symptom is uniquely indicative [1]. Usually SIBO’s most common symptoms include abdominal pain, abdominal discomfort, bloating, nausea, flatulence, and diarrhea. Moreover, its true prevalence is difficult to establish because SIBO is closely associated with various gastrointestinal diseases making it unclear whether it represents a cause or a consequence of these conditions [5]. Therefore, in the diagnostic evaluation for SIBO, the patient’s symptom profile, underlying risk factors for SIBO, and any history of previous attempts to treat other conditions need to be considered [6]. In addition, the diagnostic tools currently available, including breath tests, have limited sensitivity and specificity. Differences in diagnostic criteria, study design, and population characteristics across studies also contribute to the wide variability reported in prevalence rates [5]. Bacterial cultures from small bowel aspirate are the gold standard to diagnose SIBO. However, clinicians often treat SIBO empirically, given the cost, invasiveness, and technical difficulties in obtaining these samples [2].

The goal of SIBO treatment is to relieve the symptoms and eradicate the bacterial overgrowth, which is typically achieved by a short course of antibiotic therapy. Currently, no drugs have been approved for the treatment of SIBO and there is no specific recommendation in patients with GI cancer [7]. Actually, best available evidence supports the use of rifaximin, a nonsystemically absorbed antibiotic [8]. Studies however show significant heterogeneity regarding dose, duration of treatment, diagnostic methods, and patient population [9].

Although patients with a past or active upper GI cancer have a higher risk of developing SIBO, studies defining the role and prevalence of SIBO in this patient cohort are lacking. The aim of the present study is to assess the prevalence of SIBO and the impact of SIBO-related symptoms on quality of life in patients with a past or active upper GI cancer in a prospective study.

## 2. Materials and Methods

From April 2021 to May 2022, consecutive patients with a history of, or active upper gastrointestinal (GI) cancer who attended the oncology outpatient clinic at the University Hospital of Magdeburg were evaluated for enrollment in this prospective study. Informed consent was obtained from all individual participants included in the study. All patients presenting with abdominal pain or discomfort, bloating, nausea, flatulence, and diarrhea were initially screened and treated for other common causes of these gastrointestinal symptoms, such as steatorrhea and bile acid diarrhea. In each case, treatment was initiated or optimized according to current clinical guidelines, and symptom improvement was closely monitored. Only patients who continued to experience persistent gastrointestinal symptoms after exclusion and adequate management of other potential causes were considered eligible for inclusion. Asymptomatic patients were not enrolled. This approach focused on patients in whom SIBO was a likely clinical consideration, allowing for a more accurate characterization of symptomatic cases rather than a general prevalence estimate in all upper GI cancer patients.

Enrolled patients were asked to complete a standardized, validated questionnaire assessing the presence and severity of symptoms commonly associated with small intestinal bacterial overgrowth (SIBO). The questionnaire also documented concomitant medications (e.g., proton pump inhibitors, pancreatic enzyme supplements and opioids), relevant comorbidities, and detailed surgical history, including the type and date of previous upper GI interventions.

To quantify the overall symptom burden, patients rated the impact of SIBO-related symptoms on their quality of life using a numerical scale from 0 (no symptoms) to 3 (severe symptoms), with higher scores indicating greater impairment.

After a minimum of 30 consecutive days without antibiotic therapy, only symptomatic patients with persistent bloating, diarrhea, and impaired quality of life (score > 1) proceeded to diagnostic upper endoscopy. During the procedure, duodenal aspirates were collected from the pars descendens using a sterile, straight endoscopic retrograde cholangiography (ERC) catheter passed through a standard gastroscope (Fujifilm EG-series). All procedures were performed under propofol-based sedation to ensure patient comfort and minimize movement during endoscope insertion. Following the protocol described by Dreskin et al. [10], special care was taken to reduce contamination from oral or gastric contents. The gastroscope was advanced under direct vision to the second portion of the duodenum, and no aspiration was performed prior to reaching the distal duodenum to prevent introduction of oropharyngeal or gastric flora into the sampling channel. Duodenal fluid was collected under sterile conditions using a straight ERC catheter passed through the working channel. Approximately 1–2 mL of aspirate was obtained, immediately transferred into a sterile container, placed on ice, and transported to the microbiology laboratory for quantitative aerobic and anaerobic cultures. In accordance with the North American consensus definition, SIBO was diagnosed when bacterial counts exceeded 10^3^ colony-forming units per milliliter of duodenal aspirate.

Patients meeting the SIBO criteria were prescribed rifaximin 550 mg twice daily for seven days. To evaluate therapeutic efficacy, a follow-up questionnaire assessing the impact of SIBO-related symptoms on quality of life was administered four weeks after the completion of antibiotic therapy, ensuring sufficient time for symptom resolution or recurrence to become evident.

The study was designed as a descriptive exploratory analysis; no formal hypothesis testing was planned. An a priori plausibility check was performed assuming a SIBO prevalence of 30% versus 10% in patients with upper gastrointestinal cancer, depending on FB status. For subgroup analysis, the anticipated incidence of culture-confirmed SIBO was 80% in FB-positive and 0% in FB-negative patients. Assuming α = 0.05 and 80% power, the minimum required sample size was four patients per group.

The study was conducted in strict accordance with the ethical principles outlined in the Declaration of Helsinki [11]. The protocol, including any amendments, was reviewed and approved by the Ethics Committee of the Otto von Guericke University of Magdeburg (study number 05/21) on 15 March 2021, and all data were handled in compliance with applicable data protection regulations.

## 3. Results

A total of 90 consecutive patients with a past or active upper GI cancer (46 female, median age 65 years, range 36–87 years) were included in the study (Figure 1). The specific cancer types we observed were: 31 (34%) cases of gastric cancer, 32 (35%) patients with pancreatic cancer, 14 (17%) patients with biliary cancer, and 13 (14%) cases of esophageal cancer. Active disease was reported in 50 (55%) and past disease in 40 (45%) patients. Surgical procedures were documented in 53 (59%) of the 90 patients. In detail, pancreatic surgery was performed in 23 patients, comprising 15 Whipple procedures and 8 middle segmental pancreatic resections. Esophageal surgery was conducted in 12 patients, including 11 esophageal resections and 1 esophagectomy. Gastric surgery was performed in 10 patients, comprising 2 gastric resections and 8 gastrectomies. Furthermore, laparoscopic exploration was undertaken in 8 patients.

Systemic chemotherapy was documented in 62 (69%) of 90 patients. A completed cycle of chemotherapy was reported in 15 (24%) of 62 patients and an ongoing chemotherapy in 47 (76%) patients.

All enrolled patients (*n* = 90) completed the standardized questionnaire about SIBO-related symptoms. SIBO-related symptoms were recorded in 60 (67%) patients. The most common symptom of SIBO was a combination of bloating and diarrhea, reported in 32 (54%) of these 60 patients. Bloating alone was stated in 21 (35%) patients and diarrhea in 7 (11%) patients. Considering the impact of SIBO-related symptoms on quality of life, 42 patients (70%) reported a poor quality of life (score > 1). Detailed information on the impact of each SIBO-related symptom on quality of life is reported in Figure 2A. According to the findings in the study population, among patients with SIBO-related symptoms, 23 (39%) had gastric cancer, 23 (39%) had pancreatic cancer, 9 (14%) had biliary cancer and 5 (8%) had esophageal cancer.

An active disease was reported in 44 patients (73%) and a past disease in 16 (27%) patients with SIBO-related symptoms. 67% (40 out 60 patients) of the patients with SIBO-related symptoms were under treatment with PPI, 27% (16 out 60 patients) with opioid and 74% (44 out 60 patients) under a substitution with pancreatic enzyme.

Upper endoscopy with duodenal aspiration was performed in 46 patients but due to insufficient volume of duodenal aspirate 10 patients were excluded from the analysis. SIBO was diagnosed in 19 (53%) of 36 patients. Detailed information on patients with confirmed SIBO is reported in Table 1. The most common symptoms in patients with confirmed SIBO were bloating (95%, 18 out 19 patients) and diarrhea (68%, 13 out 19 patients). A therapy with PPI and pancreatic enzyme was present in 58% (11 out 19 patients) and 68% (13 out 19 patients) of the patients, respectively. Only 1 patient with SIBO (5%) was under therapy with opioids. Antibiotic pretreatment occurred in three patients (16%); in one case, rifaximin had been administered for prior SIBO. Although the actual definition of SIBO is a change in the number of bacteria in small bowel, the most common reported bacteria was *Escherichia coli*, followed by *Klebsiella* spp. (Figure 3).

Of the 19 patients with SIBO, 6 were treated with rifaximin 550 mg bid for 7 days. The majority of the remaining patients were excluded owing to chemotherapy-related complications, hospitalization, or the concomitant use of systemic antibiotics. Five out of six patients completed the questionnaire on the impact of SIBO-related symptoms on quality of life. Unfortunately, four weeks after the administration of antibiotics, there was no appreciable change in the impact of quality of life for bloating or diarrhea (Figure 2B). The majority of patients (4 out 6 patients) described a partial improvement immediately prior to the cessation of antibiotic therapy, followed by a rapid reversion to the pre-treatment state. One patient without symptom relief after rifaximin had previously received multiple courses of the drug for SIBO and was subsequently switched to empirical metronidazole therapy.

## 4. Discussion

To the best of our knowledge, this is the first study to assess the prevalence and impact of SIBO-related symptoms in patients with current or previous upper GI cancer. Our findings show that SIBO was present in every second patient with upper GI cancer who reported bloating or diarrhea with an impaired quality of life. Patients with SIBO were more likely to have gastric cancer (42%), active disease (79%), history of surgery (79%) and ongoing chemotherapy (58%). Notably, almost 80% of patients with SIBO had undergone gastrointestinal surgery, most commonly gastric or hepatobiliary-pancreatic procedures. It is well established that surgically induced anatomical alterations predispose to SIBO [5]. Furthermore, gastrointestinal and pancreatic resections (e.g., gastrectomy or Whipple procedure) can cause postprandial asynchrony and decreased pancreatic stimulation, leading to exocrine pancreatic insufficiency [12]. To minimize confounding due to maldigestion, only patients receiving adequate pancreatic enzyme replacement therapy were enrolled, thereby excluding clinically relevant pancreatic insufficiency as a source of gastrointestinal symptoms. A gastrectomy had been performed in 40% of our SIBO patients. After gastrectomy, the loss of gastric acid secretion allows oral bacteria to colonize the distal GI tract, resulting in a microbiota composition similar to the oral flora [13]. Another contributing factor to SIBO is PPI intake [14,15]. In our study, 58% of patients with SIBO were receiving PPI therapy. Together, surgical modification of the GI tract and PPI use appear to be the principal predisposing factors for SIBO in this population.

In our cohort, one in five patients diagnosed with SIBO had a history of cancer, although their disease was no longer active at the time of assessment. Interestingly, approximately one third of SIBO patients had never received chemotherapy, highlighting that SIBO and its associated symptom burden may develop independently of systemic anticancer therapy. These observations suggest that SIBO may contribute to persistent gastrointestinal complaints and reduced quality of life (QoL) beyond the direct effects of cancer treatment. However, interpreting QoL data in this small and heterogeneous cohort is challenging. A major limitation of our study is the small sample size, which markedly restricts the generalizability of our findings and the strength of the conclusions that can be drawn. The multifactorial nature of symptoms in patients with upper gastrointestinal cancer makes it difficult to isolate the specific contribution of SIBO to QoL impairment. Confounding factors such as chemotherapy-related toxicity, postoperative recovery, opioid or PPI use, and comorbidities may blur the clinical picture, while variability in symptom timing relative to treatment cycles or surgery adds further complexity. Given these limitations and the descriptive nature of our study, it is not possible to conclude that SIBO is the main cause of impaired QoL in this population. Nonetheless, SIBO appears to be an additional, modifiable factor that adversely affects QoL in this population. A further limitation of our study is the use of a simplified, non-validated questionnaire to assess SIBO-related symptoms and their impact on QoL. Although this pragmatic tool allowed for consistent and rapid evaluation of symptom severity (bloating, flatulence, nausea, and diarrhea) and its effect on daily life—particularly suitable for oncologic patients with fatigue and limited tolerance for long surveys—no validated instrument currently exists that specifically measures SIBO-related QoL impairment in patients with upper GI malignancies. Recently, the Small Bowel Symptom Measure (SSM) has been proposed as a dedicated SIBO-specific tool [16], but it remains in early validation and has not yet been applied in oncologic settings. Importantly, recent evidence [17] emphasizes the need to recognize SIBO as a prevalent condition that requires accurate diagnosis and individualized management. Although gas normalization is a relevant indicator of treatment response, clinical improvement and quality of life appear to depend considerably on patients’ subjective perception of their health. This perspective reinforces the importance of integrating objective and patient-reported outcomes when evaluating the impact of SIBO in individuals with upper gastrointestinal cancer. Future multicenter studies should aim to develop or employ validated SIBO-specific QoL instruments and include larger, controlled cohorts to adjust for confounders and strengthen the robustness and comparability of results.

Despite these limitations, it remains essential to recognize and address SIBO as a potentially modifiable factor contributing to QoL deterioration in cancer patients across the treatment continuum. Prospective studies using standardized, validated QoL instruments, and carefully designed to control for confounders, are urgently needed to better delineate the impact of SIBO and guide effective management strategies.

Only six of the nineteen patients diagnosed with SIBO received treatment with rifaximin. Most of the remaining patients were excluded from therapy due to complications from chemotherapy, inpatient admission, or concomitant systemic antibiotic use. The small number of treated cases limits the ability to draw firm conclusions regarding the efficacy of rifaximin in this population. All treated patients completed the symptom-specific quality of life questionnaire. Five patients reported a transient improvement in symptoms, particularly bloating and diarrhea, toward the end of the treatment period. However, this benefit dissipated within four weeks after completing the antibiotic course, with symptoms returning to baseline levels. Rifaximin appears to provide transient symptom relief but fails to correct the underlying cause of SIBO, which in most cases is anatomical alteration following surgery. These findings contrast with observations in non-oncologic populations [18], where rifaximin has demonstrated meaningful symptomatic improvement and a favorable safety profile despite moderate eradication rates. Given the limited number of treated patients and the short follow-up period, these results should be interpreted with caution. Controlled interventional studies with larger sample sizes, standardized dosing regimens, and longer follow-up are needed to better evaluate the clinical efficacy and durability of rifaximin therapy in this specific patient population.

Considering the underlying physiological context, particularly the lack of gastric acid in many of these patients, which predisposes to SIBO, future studies should explore whether alternative strategies, such as metronomic antibiotic regimens or prolonged antibiotic induction therapies, might offer more durable symptom control. Larger prospective trials are warranted to evaluate these approaches and to better define the role of rifaximin in managing SIBO among oncologic patients.

Additional studies are required to confirm our findings and develop effective therapies for this cohort of patients. Our study highlights the critical need for heightened attention to SIBO in patients with GI cancer, as addressing this condition can significantly enhance their quality of life and overall treatment outcomes.

Given the limited and transient efficacy of antibiotic therapy, SIBO should be recognized as a frequent yet often overlooked complication in patients with upper gastrointestinal malignancies, including those long after curative treatment. Persistent gastrointestinal symptoms such as bloating or diarrhea may reflect bacterial overgrowth rather than being attributable solely to prior surgery or chemotherapy. Clinicians should therefore maintain a high index of suspicion and actively evaluate symptomatic patients. While rifaximin can provide short-term relief, it rarely offers sustained benefit because the underlying anatomical alterations remain uncorrected. Management should focus on identifying and addressing predisposing factors until more effective, causal therapies become available.

## 5. Conclusions

While our study provides important insights into the prevalence and impact of SIBO in GI cancer patients, it also raises several questions that warrant further investigation. Understanding the complex interplay between cancer, surgical interventions, chemotherapy, and the gut microbiome will be crucial in developing targeted and effective strategies to prevent and treat SIBO in this vulnerable patient population.

## Figures and Tables

**Figure 1 diseases-13-00398-f001:**
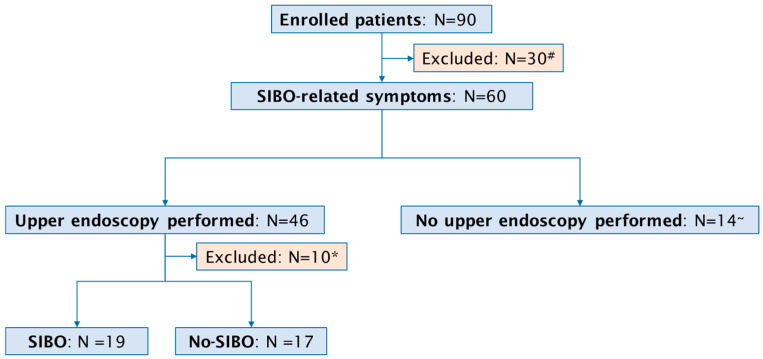
Flow chart of study population. # Absence of SIBO-related symptoms; * inadequate duodenal aspirate; ~ death before procedure (N = 7), severe infections (sepsis, cholangitis) preventing endoscopy (N = 7).

**Figure 2 diseases-13-00398-f002:**
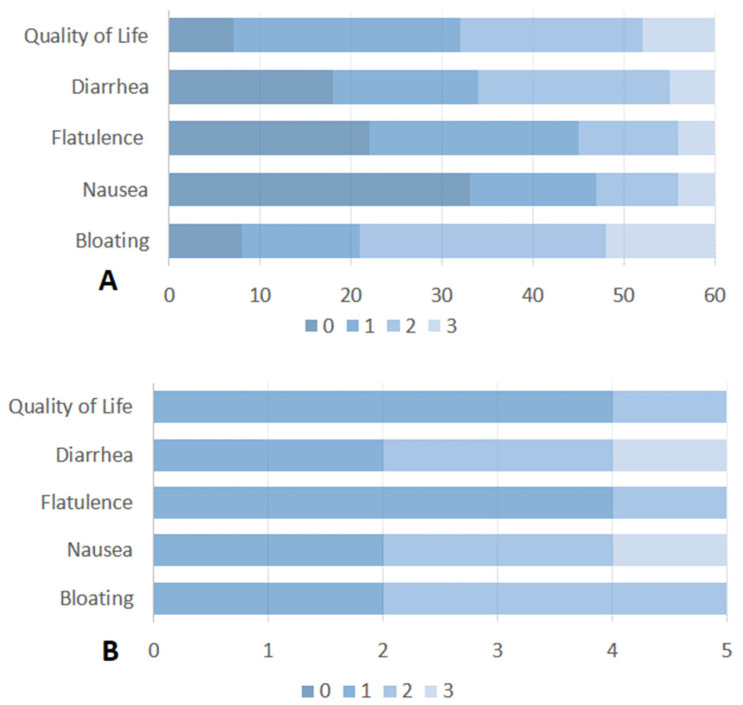
(**A**) Severity and impact of SIBO-related symptoms on quality of life of patients with active or past upper GI cancer. (**B**) Severity and impact of SIBO-related symptoms on quality of life in patients with SIBO after treatment with Rifaximin. Symptoms were rated on a scale from 0 (no symptoms) to 3 (severe symptoms).

**Figure 3 diseases-13-00398-f003:**
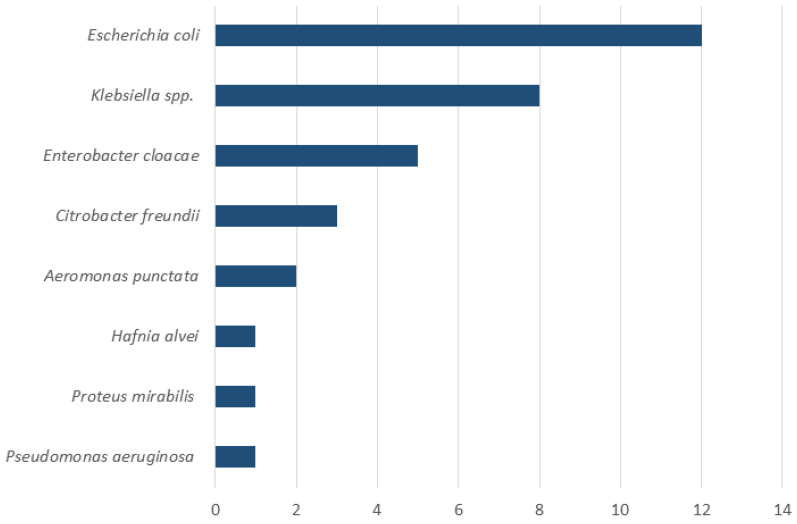
Composition of duodenal aspirates in patients with SIBO (n = 19). The *X*-axis indicates the number of patients. Note that multiple bacterial species could be isolated from the same patient.

**Table 1 diseases-13-00398-t001:** Characteristics of the 36 patients with SIBO-related symptoms who underwent upper endoscopy.

	SIBO (*n* = 19)	No SIBO (*n* = 17)
Male Gender (n, %)	10 (53%)	8 (67%)
Age (median ± range, years)	62 (33–76)	59 (37–83)
Previous Antibiotics (n, %)	3 (16%)	0
Cancer entities (n, %)		
Gastric cancer	8 (42%)	9 (53%)
Pancreatic cancer	5 (27%)	5 (29%)
Biliary tract cancer	4 (21%)	2 (12%)
Esophageal cancer	2 (10%)	1 (6%)
Stage of disease (n, %)		
Active	15 (79%)	13 (77%)
Past	4 (21%)	4 (23%)
Surgery (n, %)	15 (79%)	12 (70%)
Whipple	7 (47%)	3 (25%)
Gastric surgery	6 (40%)	8 (67%)
Biliary surgery	2 (13%)	1 (8%)
Chemotherapy		
Never	6 (31%)	3 (17%)
Completed	2 (11%)	3 (17%)
Ongoing	11 (58%)	11 (66%)

## Data Availability

The data supporting the findings of this study are not publicly available due to privacy and ethical restrictions. Requests to access the datasets should be directed to the corresponding author.

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
