# Peer review of "Prevalence and Impact on Quality of Life of Small Intestinal Bacterial Overgrowth (SIBO)-Related Symptoms in Patients with Upper Gastrointestinal Cancer"

_diseases, 2025, doi:10.3390/diseases13120398_

Round 1
Reviewer 1 Report
Comments and Suggestions for Authors
- The authors need to provide a power calculation or justification for the sample size, because the study included only 90 patients, and that small number of patients severely limits statistical power and generalizability.
- Authors need to clarify in the title, abstract, and methods that only symptomatic patients were included after exclusion of other causes. The current wording implies general prevalence among all GI cancer patients, which is, by my opinion, misleading.
- Contamination with oral flora is a known limitation of aspirate culture, but the authors did not adequately discuss this, which I believe is necessary.
- Only 6 patients received rifaximin treatment, with minimal follow-up. Therefore, I consider the conclusions about therapeutic efficacy to be unimportant with such a small number. Also, the reported “lack of sustained benefit” could be due to insufficient statistical power of the analysis, rather than real ineffectiveness. I urge the authors to address this in the article.
- A very crude symptom score of 0 to 3 was used in the quality of life assessment, which is not a validated instrument for assessing quality of life. In my opinion, this limits the strength of conclusions about patient-reported outcomes. The authors need to address this issue more seriously in the article.
- Many patients were receiving chemotherapy, proton pump inhibitors, opioids, or enzyme supplementation-all of which can strongly influence symptoms. Although some of these were described, their confounding effects were not adequately adjusted for in the analysis. The authors should conduct a multivariate analysis to control for these factors or if a multivariate analysis is not feasible due to sample size limitations, the authors should at least provide a stratified or sensitivity analysis, or explicitly discuss this as a limitation.
- The suggestion that SIBO is the main cause of impaired quality of life in this population is not sufficiently supported by the data. Therefore, the authors should distance themselves from such conclusions in their article or provide more data to support these conclusions.
- Authors need to expand the discussion to place findings in the broader context of microbiome-cancer interactions.
Reviewer 2 Report
Comments and Suggestions for Authors
The methodology of the entire article is too simplistic. The overall article lacks clear clinical value.
Reviewer 3 Report
Comments and Suggestions for Authors
Review report of manuscript “Small Intestinal Bacterial Overgrowth (Sibo) In Patients with Upper Gastrointestinal Cancer: Prevalence and Impact on Quality of Life”.
This manuscript presents a prospective study evaluating the prevalence of small intestinal bacterial overgrowth (SIBO) and its impact on symptom-related quality of life (QoL) in patients with active or previous upper gastrointestinal (GI) cancer. The topic is timely and clinically relevant, addressing an underexplored yet important complication in this patient population. The use of duodenal aspirate culture as the diagnostic gold standard strengthens the methodological rigor, and the correlation with symptom burden and treatment outcomes adds clinical depth.
Major Comments
- The study cohort is small, particularly the subgroup undergoing duodenal aspiration (n=36), limiting the generalizability of findings and statistical strength.
- Restricting inclusion to symptomatic patients may have led to an overestimation of SIBO prevalence; clarification regarding asymptomatic screening is needed.
- The QoL scoring system (0–3 scale) is simplistic and unvalidated. Use of standardized instruments (e.g., GIQLI, EORTC QLQ-C30) would improve robustness.
- Only six SIBO-positive patients received Rifaximin, and the short follow-up period limits interpretation of treatment efficacy. The discussion should further highlight the need for controlled interventional studies.
- The manuscript lacks detailed statistical analysis (e.g., p-values, confidence intervals), which is essential to support the reported associations.
Minor Comments
- The introduction could be more concise by reducing overlap in the description of SIBO pathophysiology.
- The discussion would benefit from brief practical recommendations for clinicians.
- References should be updated and standardized, including recent literature up to 2025.
Overall Assessment
A well-written and clinically meaningful study that contributes valuable preliminary data on SIBO in upper GI cancer. With minor revision, particularly enhanced statistical analysis and updated references, the manuscript would be suitable for publication.
Round 2
Reviewer 1 Report
Comments and Suggestions for Authors
I have no new comments.
